# Efficient Streaming Algorithms for Graphlet Sampling

**Yann Bourreau**[*]
Cispa Helmholtz Center for Information Security
Saarland University
Saarbrücken, Germany
yann.bourreau@cispa.de

**Marco Bressan**
Department of Computer Science
University of Milan
Milan, Italy
marco.bressan@unimi.it

**T-H. Hubert Chan**
Department of Computer Science
The University of Hong Kong
Hong Kong, China
hubert@cs.hku.hk

**Qipeng Kuang**
Department of Computer Science
The University of Hong Kong
Hong Kong, China
kuangqipeng@connect.hku.hk

**Mauro Sozio**
Institut Polytechnique de Paris, Télécom Paris
Palaiseau, France
sozio@telecom-paris.fr

## Abstract

Given a graph $G$ and a positive integer $k$, the Graphlet Sampling problem asks to sample a connected induced $k$-vertex subgraph of $G$ uniformly at random. Graphlet sampling enhances machine learning applications by transforming graph structures into feature vectors for tasks such as graph classification and subgraph identification, boosting neural network performance, and supporting clustered federated learning by capturing local structures and relationships. A recent work has shown that the problem admits an algorithm that preprocesses $G$ in time $O(nk^2 \log k + m)$, and draws one sample in expected time $k^{O(k)} \log n$, where $n = |V(G)|$ and $m = |E(G)|$. Such an algorithm relies on the assumption that the input graph fits into main memory and it does not seem to be straightforward to adapt it to very large graphs. We consider Graphlet Sampling in the semi-streaming setting, where we have a memory of $M = \Omega(n \log n)$ words, and $G$ can be only read through sequential passes over the edge list. We develop a semi-streaming algorithm that preprocesses $G$ in $p = O(\log n)$ passes and samples $\Theta(Mk^{-O(k)})$ independent uniform $k$-graphlets in $O(k)$ passes. For constant $k$, both phases run in time $O((n + m) \log n)$. We also show that the tradeoff between memory and number of passes of our algorithms is near-optimal. Our extensive evaluation on very large graphs shows the effectiveness of our algorithms.

## 1 Introduction

Sampling and counting small subgraphs in large graphs is a central problem in graph mining and machine learning. Let $G = (V, E)$ be a simple graph (where $n = |V|$ and $m = |E|$) and $k > 2$ an integer. A *k-graphlet* of $G$ is a connected, induced subgraph of $G$ on $k$ vertices. It is well known that the $k$-graphlet distribution (the relative frequency of $k$-cliques, $k$-paths, and so on) reveals key information about the structure of a graph [MSOI+02] and can be transformed into the

---

[*]Authors appear in alphabetical order.

38th Conference on Neural Information Processing Systems (NeurIPS 2024).

feature vector for graph classification and subgraph identification [TLT+19]. Graphlet information also contributes to neural network in several ways. [PLG+20] converted the graph to a matrix by sampling $k$-graphlets for a sequence of vertices so that convolutional neural networks can be applied. [TLT+19] showed that using graphlet frequency as a node attribute improves the performance of diffusion-convolutional neural networks for node classification and graph classification. It is also at the heart of graph mining applications such as graph kernels [SVP+09]. Furthermore, in clustered federated learning [GCYR22], graphlet sampling can be applied to improve model performance by capturing the local structures and characteristics of the data distributed across various clients or devices. Specifically, graphlet distributions can be used to model relationships between clients or their data in a way that reflects the underlying graph structure, allowing for more informed clustering decisions. As a consequence, sampling $k$-graphlets uniformly at random from $G$ has become a key primitive in graph mining and machine learning. This problem is called Graphlet Sampling, and many algorithms have been proposed for it [ABH19, BRRAH12, BCK+17, BCK+18, BLP19, BLP21, CLWL16, HS16, MG20, PSS19, SH15, WLR+14].

In this work we give efficient *streaming algorithms* for Graphlet Sampling. The computational model, called *streaming model*, is as follows. First, the graph $G$ can only be read by scanning its edges sequentially, in an arbitrary order that is unknown to the algorithm. Each scan of the list is called a *pass*. Second, the algorithm is only granted $M = o(|G|)$ words of memory of $\Theta(\log n)$ bits each, where $n = |V|$. For the case $M = o(n)$, the algorithm cannot even store the vertex set and is too restrictive. In this work, we consider the *semi-streaming* model, in which the algorithm has memory $M = \Omega(n)$ and the input graph is dense such that $M = o(m)$. Hence, the algorithm can store the degrees of $G$ but not the entire graph. The goal is to find an algorithm that makes a small number of passes, usually $O(\log n)$. This kind of algorithms is widely studied in graph mining, as they are useful when the input data are stored remotely or are too large to fit in main memory.

Let us discuss the problem in more detail under the standard RAM model of computation. Sampling a $k$-vertex subgraph of $G$ that is *not* necessarily connected is straightforward: just sample a $k$-vertex set $S \subseteq V$ and return $G[S]$. The problem becomes nontrivial, though, when the subgraph must be connected (i.e., a $k$-graphlet). In this case, one very general approach is *rejection sampling*: one samples some connected subgraph $G[S]$ from some distribution, and then accepts $G[S]$ with some appropriate probability, so that the resulting distribution is as close to uniform as possible. For almost a decade, all known rejection sampling techniques had a worst-case running time of $\Theta(n^k)$, or sampled from a nonuniform distribution [BRRAH12, WLR+14, SH15, BLP19, BLP21]. It was then shown that both issues can be avoided by using a simple two-phase algorithm, UGS (Uniform Graphlet Sampler) [Bre21a, Bre23]. The key idea behind UGS is to sort $G$ topologically by repeatedly deleting a vertex of maximum degree; this takes time $O(nk^2 \log k + m)$ where $m = |E|$. Afterwards, one chooses a starting vertex $v$ according to a certain distribution computed from the topological order, creates a $k$-graphlet $g$ by exploring the graph surrounding $v$ with vertices topologically no smaller than $v$ in a randomized way, and accepts $g$ with a certain probability (otherwise, the process is repeated). It can be shown that the distribution of the accepted graphlets is uniform, and that the expected time before accepting a graphlet is $k^{O(k)} \log n$.

It is therefore natural to apply UGS to the streaming setting. This is not so immediate, though. One key obstacle is that the computation of the topological order of $G$ seems inherently sequential: after deleting a maximum-degree vertex from $G$, one needs to update the degrees of its neighbors, and in the streaming model this would require one pass. We would thus make $n$ passes in total, which is unacceptable. In this work we show how to overcome these kind of obstacles. We obtain an algorithm that in $\widetilde{O}(\log n)$ passes computes an approximate topological order of $G$, and afterwards can sample $\Theta(M/k^{O(k)})$ independent uniform random graphlets every $O(k)$ passes, as long as the algorithm has $M = \Omega(kn)$ words of memory. We make several other nontrivial improvements that has practical impacts. For instance, while deciding if each vertex is contained in a sized-$k$ graphlet takes $k$ passes in a straightforward implementation, we show how to do that in only 1 pass. In the sampling phase, we show how to share computation between parallel trials such that the running time can be reduced by a factor of $M$ (and thus a factor of $n$), even though the distributions of the parallel instances are still independent. Specifically, if there are $Q$ parallel instances, instead of taking $O(mQ)$ time per pass on the edge list, we reduce the running time to $O(m \log Q)$. This requires a careful orchestration of random number generation, sorting, and prefix sum computations. We also prove that any $p$-pass streaming algorithm for Graphlet Sampling requires memory $\Omega\left(\frac{n}{p}\right)$; therefore, our algorithm is

nearly space-optimal. Finally, we conduct a series of experiments which verify the relation between several metrics and the parameters of the algorithm.

## 1.1 Contributions

**An efficient streaming algorithm for graphlet sampling.** We present STREAM-UGS, a graphlet sampling algorithm with the following formal guarantees:

**Theorem 1.1.** *In the semi-streaming model with $k = o(\log n)$ and $M$ words of memory satisfying $\Omega(n \log n) \le M = o(m)$,* STREAM-UGS *satisfies the following.[2] The preprocessing phase makes $p = O(\log n)$ passes with high probability,[3] and has running time $O(p\,m + n\,k^2 \lg k)$. The sampling phase makes $2k$ passes, returns $\Theta(M k^{-O(k)})$ independent uniform $k$-graphlets from $G$ with high probability, and runs in time $O(n\,2^k \log n + m\,k \log n)$.*

It is useful to instantiate Theorem 1.1 with $k = O(1)$. For example, $k = 3, 4, 5$ is common in the literature. In this case STREAM-UGS makes $O(\log n)$ preprocessing passes; afterwards, every $O(1)$ passes it returns a batch of $\Omega(n \log n)$ random uniform $k$-graphlets of $G$. Both preprocessing and sampling take time $O(m \log n)$. Note that the sampling phase can be repeated arbitrarily many times; every execution yields a new batch of independent uniform $k$-graphlets.

**Tradeoff Between Memory and Number of Passes.** We remark that Theorem 1.1 is a special case of a more general tradeoff that we prove between the number of passes, the memory size, and the number of graphlets returned. For example, if $M = \Omega(kn)$ then STREAM-UGS still works, but the number of preprocessing passes becomes $O(\log n \log \log n)$, or the number of graphlets sampled decreases by a factor $O\big(\log^k(n)\big)$. From a technical standpoint, STREAM-UGS relies on several ingredients. The first one is a $O(\log n)$-pass streaming algorithm for computing the said approximate topological order. The second is a 1-pass routine for computing, simultaneously from each vertex, a truncated breadth-first search to detect those vertices that do not lead to any graphlet. The third is a scheme for running instances of the sampling phase in parallel without duplicating the stream across all instances. All these results are detailed in Section 3.

**Space lower bounds for Graphlet Sampling.** We complement our guarantees by formal lower bounds on the amount of memory $M$ needed by any algorithm for Graphlet Sampling. We prove:

**Theorem 1.2.** *For $k \ge 3$, any $p$-pass streaming algorithm for Graphlet Sampling requires $\Omega\left(n/p\right)$ bits of memory.*

We also give a similar lower bound for the problem of computing the approximate topological order mentioned above. These lower bounds are obtained by suitable space-preserving reductions from the Set Disjointness, a well-known problem in communication complexity, and its multiparty variants. The full results are given in Section 4.

**Experimental evaluation.** We conduct an experimental evaluation on real-world graphs containing up to 1.8 billion edges. In particular, we study several metrics as a function of the main parameters of our algorithms. Our evaluation shows that our algorithms are efficient in practice thereby providing a valuable tool in graph mining.

## 1.2 Related Work

Most streaming algorithms for counting or sampling $k$-graphlets are specifically developed for counting or sampling triangles, and do not give guarantees for $k > 3$ [BYKS02, BBCG08, PTTW13, ADNK14, JSP15, SERU17, LJK18]. The only algorithms that count or sample $k$-graphlets for $k > 3$ with formal guarantees, [KMSS12] and [DSTU17], do not seem to guarantee uniform sampling. Their main goal is indeed to estimate the total number of occurrences of a given graphlet $H$ in the graph $G$, and they use (biased) sampling as a subroutine. It is not clear that those algorithms can be adapted to yield efficient and truly uniform $k$-graphlet sampling.

The only algorithm comparable to ours is MOTIVO [BLP19], which is based on the celebrated color-coding technique of Alon, Yuster and Zwick. MOTIVO has a preprocessing phase and a

---

[2]We need $k = o(\log n)$, otherwise the bounds become vacuous. This is not a drawback of STREAM-UGS: all known algorithms pay a factor $k^{O(k)}$ in their guarantees.

[3]In this work, "high probability" means probability $1 - n^{-c}$ where $c > 0$ is arbitrarily large but fixed.

sampling phase, and they can be implemented in $O(k)$ passes over $G$ using memory $M = \Omega(c^k\, n)$ for some $c > 0$. Crucially, however, MOTIVO does *not* yield uniform samples: the only graphlets that are sampled with nonzero probability are those made colorful by the color-coding step in the preprocessing. Our algorithm, instead, returns truly uniform samples. We also note that, in practice, MOTIVO uses memory that becomes quickly prohibitive with $k$ [BLP19], while STREAM-UGS uses memory independent of $k$.

Exploiting topological orders of $G$ is common in subgraph mining. The most used order is arguably the degeneracy order, or core order, that yields efficient subgraph-counting algorithms [MB83, CN85, Bre21b, BGL$^+$22], and for which efficient streaming algorithms exist [BKV12, SGJS$^+$13]. However, while a core order is obtained by repeatedly removing from $G$ a vertex of *minimum* degree, our topological order requires to remove a vertex of *maximum* degree. The two orders may look related at a first glance; for instance, one may be the reversal of the other. It turns out that this resemblance is only apparent, see Section 2. Thus, the algorithms of [BKV12, SGJS$^+$13] seem not useful here, and we need to develop a new one.

It should be noted that Graphlet Sampling is *not* the same problem as sampling from $G$ the occurrences of a prescribed $k$-vertex pattern graph $H$. This can be seen by letting $H$ be a $k$-vertex clique: in that case, sampling an occurrence of $H$ in $G$ solves the Clique problem, and this requires time $n^{\Omega(k)}$ unless the widely-accepted Exponential Time Hypothesis fails [CHKX06]. Graphlet Sampling instead can be solved in FPT time, that is, time $f(k) \cdot n^{O(1)}$, as [Bre21a, Bre23] and this work show.

Finally, we follow the typical technique of proving the streaming lower bound by considering the communication complexity on the famous communication game [CKS03, BJKS04]. In such proof, the game is reduced to a streaming algorithm executed by players so that a better tradeoff between memory and number of passes implies a communication cost contradicted to its lower bound. An elaboration in detail is presented in Section 4.

## 2 Preliminaries

Let $G = (V, E)$ be a simple graph. We assume $V = \{1, \ldots, n\}$. For a total order $\prec$ over $V$, we use $G(v)$ to denote the subgraph of $G$ induced by $\{u : u \succeq v\}$, i.e., the subset of vertices containing $v$ and all vertices after $v$ in $\prec$. Given $u \in V$ we let $\mathcal{N}(u|G(v)) = \{w \succeq v : \{u, w\} \in E\}$ be the neighborhood of $u$ in $G(v)$, and $d(u|G(v)) = |\mathcal{N}(u|G(v))|$ be the degree of $u$ in $G(v)$. We denote by $\Delta$ the maximum degree of $G$. The following notion will be crucial:

**Definition 2.1** ($\vartheta$-DD Order). *Let $\vartheta \in (0, 1]$. A total order $\prec$ over $V$ is a $\vartheta$-degree-dominating order ($\vartheta$-DD order) if, for all $v \in V$ with $d(v|G(v)) > 1$ and all $u \succeq v$, we have $d(v|G(v)) \geq \vartheta \cdot d(u|G(v))$.*

The topological order used by UGS [Bre23] is a $1$-DD order. A $\vartheta$-DD order is an approximation of a $1$-DD order. Using a $\vartheta$-DD order yields the same guarantees of UGS, only with an acceptance probability scaled by $\vartheta^{k-1}$ (see Lemma C.1).

**Computational model.** The algorithm has a memory of $M$ words of $\Theta(\log n)$ bits each. We require $M = \Omega(n)$ or $M = \Omega(n \log n)$, depending on the case. The graph is stored as a list of edges in arbitrary (i.e., adversarial) order. With one *pass*, the algorithm can read the list sequentially. This is called *semi-streaming* model [FKM$^+$05]. To avoid trivialities we assume $m = \Omega(M)$; otherwise, in one pass one can store $G$ and run the algorithm of [Bre23]. For computation we assume the standard RAM model. We also assume that in time $O(1)$ one can draw a random uniform integer in $\{1, \ldots, c\}$ for any $c = \text{poly}(n)$, or a Bernoulli random variable $B(p)$ for $p = \Omega(n^{-k})$. When we run multiple instances of a subroutine in parallel, the running time is understood to be the total number of operations executed by all those instances.

**Graphlet size.** Our algorithms are designed primarily for $k = O(1)$, but they yield nontrivial guarantees also for $k = \omega(1)$; for instance, for $k = \sqrt{\log n}$. Our full statements make the dependence on $k$ clear.

**DD orders vs. core orders**. It should be noted that $1$-DD orders and core orders do not seem related in any useful way. For example, let $G$ be the disjoint union of a star with $\delta_1$ leaves and a $\delta_2$-clique. If $\delta_1 > \delta_2$, then the center of the star comes first in a $1$-DD order, but in a core order it sits after the

star's leaves and before the clique vertices. If $\delta_2 > \delta_1$, instead, then in a core order the vertices of the star all come before the clique, while in a 1-DD ordering the center of the star sits after $\delta_2 - \delta_1$ vertices of the clique.

# 3 A Streaming-Based Graphlet Sampling Algorithm

This section describes our semi-streaming algorithm, STREAM-UGS. STREAM-UGS is based on the approach of UGS [Bre23], and runs in two phases: preprocessing and sampling.

**Preprocessing.** This is done once, and involves two steps.

P1 Computing a $\vartheta$-DD order $\prec$ of $G$. The order defines implicitly the subgraphs $G(v)$, see Section 2, as well as a certain distribution $\vec{p}$ over $V$ that is needed by the sampling phase, see below.

P2 Computing the said probability distribution $\vec{p}$ on $V$. Let $\mathcal{G}(v)$ be the set of all graphlets that are subgraphs of $G(v)$ and contain $v$. This is called the *bucket* of $v$. The distribution $\vec{p}$ can be seen as a distribution over the buckets $\mathcal{G}(v)$. For every $v \in V$, the probability $p(v)$ is proportional to $d(v|G(v))^{k-1}$ if $\mathcal{G}(v) \neq \emptyset$, and $p(v) = 0$ otherwise. The entry $p(v)$ will be the probability that $v$ is picked in the first step of the subsequent sampling.

**Sampling.** This phase may report failure, but if it succeeds, it returns a uniformly random $k$-graphlet of $G$. The expected number of trials to successfully return a graphlet is $\vartheta^{1-k} \cdot k^{O(k)}$. However, with enough memory, one can carry out independent trials in parallel. The phase consists of three steps.

S1 Choosing a bucket $\mathcal{G}(v)$ according to the probability distribution $\vec{p}$. One can show that, if $\mathcal{G}(v) \neq \emptyset$, then $|\mathcal{G}(v)|$ equals $d(v|G(v))^{k-1}$, up to factors $k^{O(k)}$. Thus, we are choosing buckets with probability roughly proportional to the number of $k$-graphlets in them.

S2 Sampling a random graphlet $G[S] \in \mathcal{G}(v)$. To this end let $S_1 = \{v\}$. For $i = 1, 2, \ldots, k-1$, draw an edge uniformly at random from the cut between $S_i$ and $G(v)$, and add the endpoint to $S_i$; this gives $S_{i+1}$. The final set $S = S_k$ defines the graphlet $G[S]$. Note that the sampling here is not uniform. Uniformity is ensured by the next step.

S3 Accepting $S$ with some probability $p(S)$. If $S$ is accepted then compute $G[S]$ and return it; otherwise, report failure.

**Technical Challenges.** In the semi-streaming setting, we cannot store all the edges of $G$ in the memory. Our contributions are how to implement each of the above steps with restricted memory using a small number of passes on the edge list.

## 3.1 Computing a $\vartheta$-DD order

We present an algorithm, APPROX-DD (Algorithm 1), that given a graph $G$ and a parameter $0 < \epsilon = O(1)$, computes a $\vartheta$-DD order of $G$, where $\vartheta = \frac{1}{1+\epsilon}$. Our main result is:

**Theorem 3.1.** *In the semi-streaming model,* APPROX-DD *can be implemented so that with probability at least* $1 - n^{-\Theta(1)}$*, it makes* $q = O\left(\frac{\log n}{\epsilon^2} \log \frac{\log n}{\epsilon}\right)$ *passes if* $M = \Omega(n)$*, and* $q = O\left(\frac{\log n}{\epsilon^2}\right)$ *passes if* $M = \Omega\left(\frac{n \log n}{\epsilon}\right)$*. In both cases the total running time is in* $O(q \cdot m)$*.*

The complete proof of Theorem 3.1 combines several technical lemmas and can be found in Appendix A. The rest of this section overviews APPROX-DD and sketches the proof. APPROX-DD maintains a list $L$, initially empty, which will eventually contain the vertices of $V$ sorted by $\prec$. The algorithm proceeds in *peeling* rounds; every round deletes from $G$ all vertices of degree at least $\frac{\Delta}{1+\alpha}$, for some appropriate $\alpha$, and places them in $L$ in the right order. This "right order" is not simply any order; it is carefully computed by another procedure, called *shaving*, using random partitioning. Algorithm 1 gives the pseudocode of APPROX-DD. It assumes that $\delta_G(\cdot)$, and all other degrees, are updated as soon as the graphs are modified; except for $\Delta$, which is explicitly recomputed when needed. In practice, updating the degrees amounts to running one pass over the edges of $G$. The algorithm has a triply nested loop structure, and we use the subroutines PEEL and SHAVE for clearer illustration.

---

**Algorithm 1** Approximate DD Order. The lines marked with a ▷ entail a recomputation of the degrees and, thus, one streaming pass.

---

1: **function** APPROX-DD$(G, \epsilon)$
2:     $L \leftarrow$ empty list
3:     **while** $V(G) \neq \emptyset$ **do**
4:         $(G, L) \leftarrow$ PEEL$(G, \epsilon, L)$
5:     **return** $L$

6: **function** PEEL$(G, \epsilon, L)$
7:     $\Delta \leftarrow \max_{v \in V(G)} d(v|G)$                                    ▷
8:     $\alpha \leftarrow \frac{\epsilon}{2}$
9:     $H \leftarrow \{v \in V(G) : d(v|G) \geq \frac{\Delta}{1+\alpha}\}$
10:     **while** $H \neq \emptyset$ **do**
11:         $(G, L, H) \leftarrow$ SHAVE$(G, \epsilon, \Delta, L, H)$
12:     **return** $(G, L)$

13: **function** SHAVE$(G, \epsilon, \Delta, L, H)$
14:     $\beta \leftarrow \frac{2+2\epsilon+\epsilon^2}{\epsilon}, \ell \leftarrow \lceil 2(1+\beta) \rceil$
15:     let $H_1, \ldots, H_\ell$ be a uniform random partition of $H$
16:     let $R_i \leftarrow \{v \in H_i : d(v|G[H_i]) \leq \frac{\Delta}{1+\beta}\}$ for $i \in [\ell]$       ▷
17:     **for** $i = 1, \ldots, \ell$ **do**
18:         $S_i \leftarrow \{v \in R_i : d(v|G) \geq \frac{\Delta}{1+\alpha}\}$
19:         append the vertices of $S_i$ to $L$ in arbitrary order
20:         $G \leftarrow G \setminus S_i$                                          ▷
21:     $H \leftarrow H \setminus \cup_{i \in [\ell]} R_i$
22:     **return** $(G, L, H)$

---

**Intuition for** PEEL **and** SHAVE. Let $H$ be the subset of vertices with degree at least $\frac{\Delta}{1+\alpha}$ in $G$ (line 9). As a first attempt, we could remove all vertices in $H$ from $G$ in a single batch, which is appended to $L$ in arbitrary order. This works if every vertex $v$ has all of its neighbors come after itself in $L$, since in that case $d(v|G(v)) \geq \frac{1}{1+\alpha}\Delta \geq \frac{1}{1+\epsilon}d(u|G(v))$ for all $u \succeq v$. However, in general this does not happen.

To this end, we attempt to remove vertices in $H$ from $G$ in phases through SHAVE. Specifically, we assign each vertex in $H$ to one of $\ell = \lceil 2(1+\beta) \rceil$ batches $H_1, \ldots, H_\ell$ independently and uniformly at random. By Markov's inequality, each vertex has at most $\frac{\Delta}{1+\beta}$ neighbors within $H_i$ with good probability; then, in phase $i$, every such vertex in $H_i$ that has current degree at least $\frac{\Delta}{1+\alpha}$ will be removed together in one batch from $G$ (and appended to $L$ in an arbitrary order). The degrees of the remaining vertices in $G$ will then be updated. Each call to SHAVE has $\ell$ such phases.

The procedure performed by SHAVE does not yet remove $H$ from $G$; it removes only the vertices that have few neighbors in the same subset $H_i$. To completely remove $H$, we need to iterate the procedure. This is what PEEL does at line 10. We shall see that there are two cases. When $\Delta$ is large enough, then one call to SHAVE in line 11 will be sufficient to make $H$ empty with high probability. When $\Delta$ is small, we may instead need $O(\log n)$ calls to SHAVE in order for $H$ to become empty. This is, technically speaking, what makes the whole algorithm APPROX-DD efficient. Formally, we prove the following (in Appendix A):

**Lemma 3.2.** *Let $p \in (0, 1)$. With probability at least $1 - p$, one execution of* PEEL *(Algorithm 1) calls* SHAVE $O\left(\log \frac{n}{p}\right)$ *times, and at most once if $\Delta \geq 12(1+\alpha)(1+\beta) \ln \frac{n}{p}$.*

## 3.2 Finding the Non-Empty Buckets

Algorithm 2 gives the pseudocode of COMPUTEDISTRIB, which computes the distribution $\vec{p}$ and the normalization constant $\Gamma$ needed by the sampling phase. As said above, the crux of COMPUTEDISTRIB is checking if $\mathcal{G}(v) \neq \emptyset$ for each $v \in V$. To this end, it suffices to perform a BFS on all vertices in parallel, so as to check whether the connected component of $v$ in $G(v)$ has size at least $k$. A naive method would take $k$ passes on the edge list, but with $\Omega(kn)$ words of memory, we can achieve this in 1 passes, as Lemma 3.3 proves.

---

**Algorithm 2** COMPUTEDISTRIB

---

1: **function** COMPUTEDISTRIB($\prec$)
2:     **for** each $v \in V$ **do**
3:         compute $d(v|G(v))$ and $\mathbb{1}_v := \mathbb{1}_{\mathcal{G}(v) \neq \emptyset}$
4:         set $b_v \leftarrow \mathbb{1}_v \cdot d(v|G(v))^{k-1}$
5:     set $Z \leftarrow \sum_{v \in V} b_v$
6:     compute probability vector $\vec{p}$: for each $v \in V$, $p(v) \leftarrow \frac{b_v}{Z}$
7:     set $\Gamma \leftarrow \frac{1}{(k-1)! \, Z \, (1+\epsilon)^{k-1}}$
8:     **return** $(\vec{p}, \Gamma)$

---

**Lemma 3.3.** *In the semi-streaming model with $M = \Omega(kn)$, the routine* COMPUTEDISTRIB *(Algorithm 2) can be implemented so as to use* 1 *passes and time* $O(m + n \cdot k^2 \log k)$.

## 3.3 Sampling Graphlets in Parallel

Algorithm 3 gives the pseudocode of SAMPLE, the sampling phase of STREAM-UGS. The sampling phase works after invoking APPROX-DD and COMPUTEDISTRIB in turn (see the subroutine PREPROCESS in Algorithm 4, Appendix A). Note that Algorithm 3 gives only an abstract description of the routines. The concrete implementation details are rather involved, though, and we give them in the rest of the section. The reason for this complexity is that, since each sampling trial requires memory $O(k^2)$, then with a memory of $M$ words we can perform $\Theta(M/k^2)$ independent sampling trials in parallel. Doing so efficiently, though, is significantly less straightforward than one would think. However, we prove:

**Theorem 3.4.** *In the semi-streaming model with $M = \Omega(kn)$ words of memory, suppose the preprocessing routine* PREPROCESS *has been run with $\epsilon = 1/2$. Then, $Q = \Theta(M/k^2)$ parallel instances of* SAMPLE *can be run in a batch so that $Q/k^{O(k)} = \Theta(M/k^{O(k)})$ independent uniform random $k$-graphlets of $G$ are returned successfully with high probability. Moreover each batch can be implemented in two ways:*

- *The first one takes $k$ passes and $O(Mm)$ time.*
- *The second one takes $2k - 1$ passes and $O(M2^k + km \log n)$ time.*

*Proof Intuition.* Each instance of SAMPLE requires only $O(k^2)$ words of memory. Hence, with memory $M$, we can run $Q = \Theta(M/k^2)$ parallel instances of SAMPLE. By Lemma C.1, each instance independently returns a uniform random $k$-graphlet with probability at least $\frac{k^{-O(k)}}{(1+\epsilon)^{k-1}} = k^{-O(k)}$. By standard concentration bounds, then, with high probability the number of uniform random $k$-graphlets returned overall is $\Theta(M/k^{O(k)})$. The details of the subroutines and data structures to achieve Theorem 3.4 are given in Appendix C.

# 4 Space Lower Bounds

We show that the tradeoff between memory and number of passes of our algorithms is near-optimal. To this end, we use well known results from communication complexity. In the *multi-party randomized communication model* [CKS03, BJKS04], there are $t \geq 2$ players seeking to compute a function $f : \mathcal{X}_1 \times \mathcal{X}_2 \cdots \times \mathcal{X}_t \to \mathcal{Y}$ while exchanging messages according to a specified protocol $\Pi$. For each $i \in [t]$, the $i$-th player has an input $x_i \in \mathcal{X}_i$. Given $x_1, \ldots, x_t$, the random variable $\Pi(x_1, \ldots, x_t)$ denotes the message transcript (containing all messages) obtained when all players follow $\Pi$ on

---

**Algorithm 3** SAMPLE (Abstract Description)

---

1: **function** SAMPLE($\prec, \vec{p}, \Gamma$)                       ▷ $\prec$ is used to define $G(v)$.
2:     Sample $v$ from the distribution $\vec{p}$.
3:     $S \leftarrow$ RAND-GROW$(v, \prec)$
4:     $q_S \leftarrow$ PROB$(v, S, \prec)$
5:     With probability $\frac{\Gamma}{p(v) \cdot q_S}$, **return** $S$;
6:     Otherwise, **return** "failure"

7: **function** RAND-GROW$(v, \prec)$
8:     $S_1 \leftarrow \{v\}$
9:     **for** $i = 1, \ldots, k-1$ **do**
10:        let $E_i$ be the subset of edges in $G(v)$ with exactly one endpoint in $S_i$.
11:        sample an edge $\{u, u'\}$ uniformly at random from $E_i$, where $u \in S_i$ and $u' \notin S_i$.
12:        $S_{i+1} \leftarrow S_i \cup \{u'\}$
13:     **return** $S_k$

14: **function** PROB$(v, S, \prec)$
15:     **for** $i = 1, \ldots, k$ **do**
16:        $q_i \leftarrow$ an empty map from $\binom{S}{i}$ to $\mathbb{R}$ with default value 0
17:     $q_1(\{v\}) \leftarrow 1$
18:     **for** $i = 2, \ldots, k$ **do**
19:        **for** each $S_i \in \binom{S}{i}$ and each $w \in S_i$ **do**
20:           $n_i \leftarrow$ number of neighbors of $w$ in $S_i$
21:           $c_i \leftarrow \sum_{z \in S_i \setminus \{w\}} (d(z|G(v)) - d(z|G[S_i \setminus \{w\}]))$
22:           $q_i(S_i) \leftarrow q_i(S_i) + q_{i-1}(S_i \setminus \{w\}) \cdot \frac{n_i}{c_i}$
23:     **return** $q_k(S)$

---

inputs $x_1, \ldots, x_t$, where the randomness is over the coins of the players. $\Pi$ is a $\delta$-error protocol for $f$ if there is a deterministic function $\Pi_{\text{out}}$ such that, for all inputs $x_1, \ldots, x_t$,

$$\Pr\left(\Pi_{\text{out}}(\Pi(x_1, \ldots, x_t)) = f(x_1, \ldots, x_t)\right) \geq 1 - \delta \tag{1}$$

The communication cost of $\Pi$, denoted by $|\Pi|$, is the maximum length of $\Pi(x_1, \ldots, x_t)$ over all $x_1, \ldots, x_t$ and over all random choices of all players. The $\delta$-error randomized communication complexity of $f$ is the cost of the best $\delta$-error protocol for $f$. In the *one-way* variant of such a model, the $i$th player sends exactly one message to the $(i+1)$th player, throughout the protocol (we let $t + 1 = 1$).

In the classical *disjointness problem*, the $i$th player has an $n$-bit vector $(x_{i_1}, \ldots, x_{i_n}) \in \{0, 1\}^n$ that corresponds to a subset of a sized-$n$ ground set. The goal is to determine whether there is a common element among the subsets of all players, i.e. whether there is an index $j$ such that $x_{i_j} = 1$, for all $i \in [t]$. It is known that this problem has $\Omega(n/t)$ communication complexity in the one-way multi-party randomized communication model (Theorem 2.6 of [CKS03]). This holds even in the case when either the bit vectors represent pairwise disjoint sets or they have a unique common element but are otherwise pairwise disjoint.

We show that any $q$-pass streaming algorithm requires memory $M = \Omega(n/q)$ even for the problem of computing a node with approximate maximum degree, that is, to compute a node with rank 1 in an approximate DD-order. We believe such a result is well-known, however, we could not find a reference for a general $q$-pass algorithm. For completeness, we provide the full proofs in Appendix D.

**Lemma 4.1.** *Given a graph $G = (V, E)$, $\alpha \in (0, 1)$, any $q$-pass streaming algorithm computing a node $v$ such that $\delta_G(v) \geq \alpha \max_{w \in V} \delta_G(w)$ with probability at least $\frac{2}{3}$, requires memory $\Omega\left(\frac{\alpha^2 |V|}{q}\right)$.*

Finally, we provide a lower bound on the memory requirement of any streaming algorithm for uniform $k$-graphlet sampling, $k \geq 3$.

**Lemma 4.2.** *Given a graph $G = (V, E)$, $k \geq 3$, any $q$-pass streaming algorithm for the uniform $k$-graphlet sampling problem on $G$ requires memory $\Omega\left(\frac{|V|}{q}\right)$.*

| Dataset | File Size (MB) | #Vertices | #Edges |
|---|---|---|---|
| Dense | 1,858 | 20,000 | 159,993,472 |
| NY Times [Kun13] | 858 | 401,388 | 69,654,798 |
| Twitter (WWW) [Kun13, KLPM10] | 20,437 | 41,652,230 | 1,202,513,047 |
| Twitter (MPI) [Kun13, CHBG10] | 25,590 | 52,579,682 | 1,614,106,188 |
| Friendster [Kun13] | 32,300 | 68,349,466 | 1,811,849,343 |

Table 1: Dataset statistics. Dense is generated synthetically by drawing each edge with probability 0.8, and other four datasets are from KONECT.

## 5 Experiments

We implemented our algorithms in Python 3.9 and conducted experiments on a Ubuntu server with CPU Intel Xeon Silver 4108 (1.80GHz) and 28GB RAM. The implementation can be found in the online repository. [4]

**Datasets.** We run our experiments on several real-world graphs from the KONECT website [Kun13] [5] and a synthetic random dense graph generated by drawing each edge with probability 0.8 (we name it Dense). We removed edge directions, weights, self-loops, duplicate edges and any other irrelevant data, so as to retain only a list of undirected edges. Table 1 reports all dataset statistics after such a preprocessing phase.

**Streaming model.** The stream of edges comes from a CSV file stored in the disk representing the edge list. Our algorithms make multiple sequential passes over the input data while never storing the set of edges in main memory. Note that from Theorem 3.4 we have two approaches to do sampling in parallel, among which we implemented the second one using $2k - 1$ passes and $O(M2^k + km \log n)$ time.

**Methodology.** We constrain the memory $M$ by restricting the maximum number of edges that we can store during the computation of the DD order and sampling to be $\frac{n}{2}$ in all cases. In the sampling phase it is required to sample at least 100 graphlets successfully in order to reduce the error of sampling probability. We study several metrics as a function of $\epsilon$ and $k$, such as the number of passes, the memory usage, as well as the number of sampling trials to obtain 100 graphlets.

**A Heuristic Approach for PEEL.** We develop a heuristic approach called APPROX-DD-HEURISTIC in Appendix E to speed up the running time of Algorithm 1. We evaluate APPROX-DD-HEURISTIC on large real-world graphs to show the scalability of our algorithms and then conduct a comparison between APPROX-DD and APPROX-DD-HEURISTIC on smaller graphs.

APPROX-DD **versus** APPROX-DD-HEURISTIC**.** We conduct the comparison between APPROX-DD and APPROX-DD-HEURISTIC on NY Times, the real-world sparse graph and Dense, the synthetic dense graph. Figure 1 shows the number of passes and memory usage as a function of $\epsilon$.

In Figure 1a and 1b we can see that APPROX-DD-HEURISTIC delivers significantly better results for sparse graphs but has no significant advantage for dense graphs. This is expected, as PEEL-HEURISTIC can process at each step a relatively large chunk of the input graph, if such a graph is sparse. Moreover, consistently with our theoretical analysis in Section 3, we can observe that the number of passes for preprocessing decreases as a function of $\epsilon$, while the number of passes for sampling increases. Concerning APPROX-DD-HEURISTIC, both the number of passes and memory usage are less sensitive to $\epsilon$ since in APPROX-DD-HEURISTIC the amount of available memory plays a more important role.

As observed, APPROX-DD-HEURISTIC outperforms APPROX-DD on real-world sparse graphs. Hence, we will focus our attention on APPROX-DD-HEURISTIC in the following experiment.

**Running on Large Graphs.** Figure 2 shows several metrics as a function of $\epsilon$ and $k$ over real-world large graphs. Figure 2a and 2b show that around 30 passes are sufficient to complete the whole sampling task, while the memory usage is indeed less than the total size of the edge list. Figure 2c and 2d show the number of trials to obtain 100 graphlets as a function of $\epsilon$ and $k$, respectively. Observe that the success probability is significantly higher than the worst-case bound provided by

---

[4] https://github.com/l2l7l9p/UGS-Streaming
[5] http://konect.cc/networks/

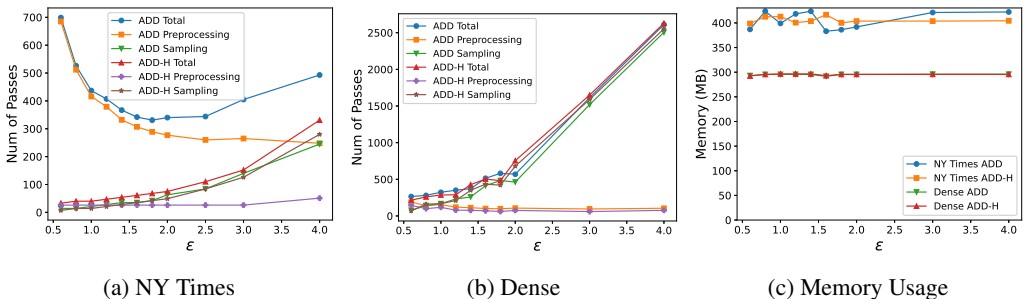

| (a) NY Times | (b) Dense | (c) Memory Usage |

Figure 1: The number of passes versus $\epsilon$ on NY Times (1a), on Dense (1b), and the memory versus $\epsilon$ on two datasets (1c), fixing $k = 4$. ADD stands for APPROX-DD and ADD-H stands for APPROX-DD-HEURISTIC.

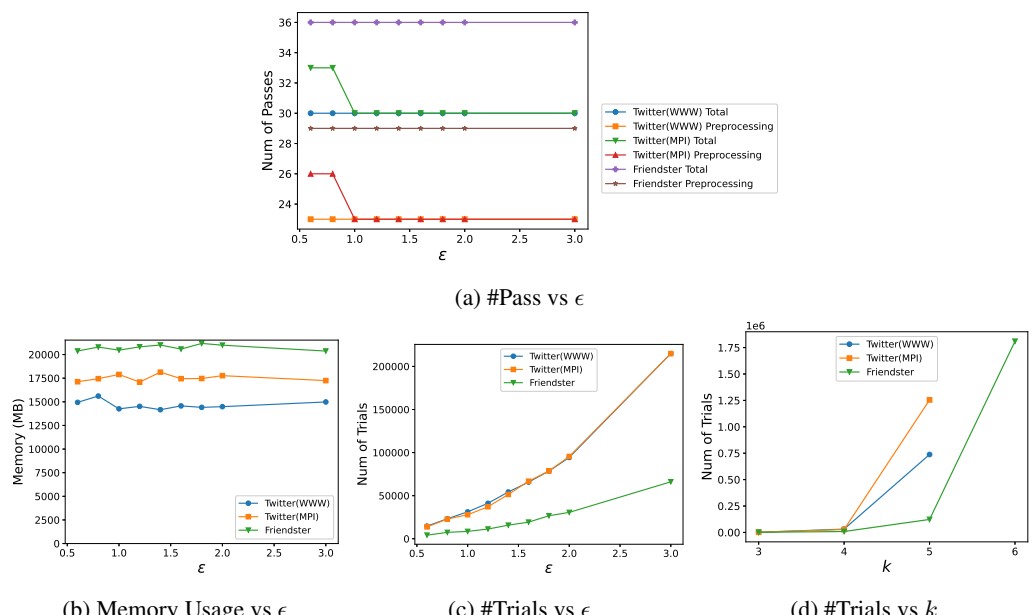

(a) #Pass vs $\epsilon$

(b) Memory Usage vs $\epsilon$      (c) #Trials vs $\epsilon$      (d) #Trials vs $k$

Figure 2: The number of passes versus $\epsilon$ (2a), the memory usage versus $\epsilon$ (2b), the number of trials needed for 100 samples versus $\epsilon$ (2c) and versus $k$ (2d) on Twitter(WWW), Twitter(MPI) and Friendster, fixing $k = 4$ in 2a, 2c, 2b and $\epsilon = 1$ in 2d.

Lemma C.1. For example, the actual success probability in Twitter(WWW) with $\epsilon = 1, k = 4$ is $100/31230 \approx 0.0032$, which is significantly larger than $9.65 \times 10^{-5}$ provided by Lemma C.1.

## 6 Conclusion and Future Work

We develop efficient semi-streaming algorithms for uniform graphlet sampling, requiring sublinear amount of memory and polylogarithmic number of passes in the size of the input data. We also provide a space lower bound showing that the tradeoff between memory and number of passes of our algorithms is near-optimal. Our theoretical results are complemented with an experimental evaluation on large real-world graphs showing the effectiveness of our algorithms and that they can provide a valuable tool in graph mining and data analysis. Our work paves the way for developing efficient algorithms in other computational model for large-scale processing, such as the MapReduce model.

## Acknowledgement

T-H. Hubert Chan was partially supported by the Hong Kong RGC grants 17201823 and 17203122.

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

# A    Proofs for Section 3.1

**Lemma A.1.** APPROX-DD$(G, \epsilon)$ *computes a* $\frac{1}{1+\epsilon}$-*DD ordering of* $G$.

*Proof.* First, consider SHAVE. We claim that, just after the execution of line 18, every vertex in $S_i$ has at least $\frac{\Delta}{1+\alpha} - \frac{\Delta}{1+\beta}$ and at most $\Delta$ neighbors in $V(G) \setminus S_i$. Consider indeed any $v \in S_i$ just after line 18 is executed. Note that $v$ has by construction at least $\frac{\Delta}{1+\alpha}$ neighbors in $V(G)$, and since $S_i \subseteq R_i$, it has at most $\frac{\Delta}{1+\beta}$ neighbors in $S_i$. The lower bound follows. For the upper bound just note that $v$ has degree at most $\Delta$ in $V(G)$ and, thus, in $V(G) \setminus S_i$.

Now consider APPROX-DD$(G, \epsilon)$. Note that the final list $L$ is obtained through the subroutine SHAVE by repeatedly computing some $S_i \subseteq V(G)$, appending $S_i$ to $L$, and deleting $S_i$ from $G$ (lines 20 and 19). On the one hand this implies that the returned list $L$ is a permutation of $V(G)$ (i.e., every vertex of $V(G)$ appears exactly once in $L$). On the other hand consider any vertex $v \in V(G)$, and let $\Delta$ be as in the subroutine PEEL when $v$ was appended to $L$. By the arguments above and by the choice of $\alpha, \beta$:

$$d(v|G(v)) \geq \frac{\Delta}{1+\alpha} - \frac{\Delta}{1+\beta} \geq \frac{\Delta}{1+\epsilon} \tag{2}$$

where $G(v)$ is the subgraph of $G$ induced by $v$ and all vertices appearing after $v$ in $L$. Moreover note that $\Delta$ is an upper bound on $d(u|G(v))$ for every $u \in G(v)$. It follows that $d(v|G(v)) \geq \frac{d(u|G(v))}{1+\epsilon}$ for all $v$ and all $u \succ v$ where $\succ$ is the ordering defined by $L$. $\square$

**Lemma A.2.** *Each call to* SHAVE *requires* $\ell + 1 = O(\epsilon + \frac{1}{\epsilon})$ *invocations of* COMPUTEALLDEG.

*Proof.* Line 16 requires, for each $i \in \ell$, one call to COMPUTEALLDEG on $G[H_i]$. For each one of the $\ell$ iterations of the loop at line 17, a call to COMPUTEALLDEG is needed after $G$ is updated at line 20. Noting that $\ell = O(\epsilon + \frac{1}{\epsilon})$ concludes the proof. $\square$

**Lemma 3.2.** *Let* $p \in (0,1)$. *With probability at least* $1 - p$, *one execution of* PEEL *(Algorithm 1) calls* SHAVE $O\left(\log \frac{n}{p}\right)$ *times, and at most once if* $\Delta \geq 12(1+\alpha)(1+\beta) \ln \frac{n}{p}$.

*Proof.* Consider a generic iteration of the loop of line 10. Fix any $v \in H_i$ and write $d_{H_i}(v)$ as:

$$d(v|H_i) = \sum_{\{u,v\} \in E(G_i)} X_i(u) \tag{3}$$

where $X_i(u)$ is the indicator variable of the event $u \in H_i$. Note that $\mathbb{E}[X_i(u)] = \frac{1}{\ell}$ and therefore

$$\mathbb{E}[d(v|H_i)] = \frac{d(v|G)}{\ell} \leq \frac{\Delta}{\ell} \leq \frac{\Delta}{2(1+\beta)} \tag{4}$$

By Markov's inequality, then, $\Pr\left(d(v|H_i) > \frac{\Delta}{1+\beta}\right) < \frac{1}{2}$. It follows that

$$\mathbb{E}[|R_i|] \geq \frac{|H_i|}{2} \tag{5}$$

and therefore

$$\mathbb{E}[|\cup_{i=1}^{\ell} R_i|] \geq \frac{|H|}{2} \tag{6}$$

By standard arguments this implies that, for all $p > 0$, with probability at least $1 - p$ after $O(\log \frac{n}{p})$ iterations of the the loop of line 10 the set $H$ becomes empty and thus the loop terminates.

For the second claim, note that $\ell \leq 4(1+\beta)$ and $d(v|G) \geq \frac{\Delta}{1+\alpha}$ imply

$$\mathbb{E}[d(v|H_i)] \geq \frac{\Delta}{4(1+\alpha)(1+\beta)} \tag{7}$$

Now consider again (3), and note that the random variables

$$\{X_i(u)\}_{\{u,v\} \in E(G)} \tag{8}$$

are independent by construction of $H_i$. By standard concentration bounds then:

$$\Pr\left(d(v|H_i) > \frac{\Delta}{(1+\beta)}\right) \le \Pr\left(d(v|H_i) > 2 \cdot \mathbb{E}[d(v|H_i)]\right) \tag{9}$$

$$\le \exp\left(-\frac{\mathbb{E}[d(v|H_i)]}{3}\right) \tag{10}$$

$$\le \exp\left(-\frac{\Delta}{12(1+\alpha)(1+\beta)}\right) \tag{11}$$

Thus for $\Delta \ge 12(1+\alpha)(1+\beta)\ln\frac{n}{p}$ we have $\Pr\left(d(v|H_i) > \frac{\Delta}{(1+\beta)}\right) \le \frac{1}{n}$. By a union bound over all $v \in H$ this implies that, with probability at least $1 - p$, for all $i \in [\ell]$ we have $R_i = H_i$. If that is the case then $H$ becomes empty when line 21 is executed, proving the claim. $\qquad\square$

**Lemma A.3.** *When the subroutine* PEEL *is called on a graph $G$ with maximum degree $\Delta$, it removes vertices from $G$ so that, upon termination, $G$ has maximum degree smaller than $\frac{\Delta}{1+\alpha}$. Moreover, if $\Delta < 1$, then* PEEL *terminates after one iteration of the loop of line 10 and returns $G = \emptyset$.*

*Proof.* Denote by $G'$ the value of $G$ when the loop of line 10 ends. Suppose $G'$ contains a vertex $v$ such that $d_{G'}(v) \ge \frac{\Delta}{1+\alpha}$. Since $G \supseteq G'$ then $d_G(v) \ge \frac{\Delta}{1+\alpha}$, and thus $v \in H$ by line 9; and since the loop ends when $H = \emptyset$, then at some iteration $v \in R_i$ for some $i \in [\ell]$. Consider then that iteration and let $G$ be the current graph; obviously $G \supseteq G'$. If $v \in S_i$ then by line 20 $v \notin G$ and thus $v \notin G'$. If instead $v \notin S_i$ then by line 18 $d_{G'}(v) \le d_G(v) < \frac{\Delta}{1+\alpha}$. Both cases give a contradiction.

For the second claim note that if $\Delta < 1$ then $\Delta = 0$. In that case $H_i = R_i = S_i$ for all $i \in [\ell]$, and the claim follows by construction of the algorithm. $\qquad\square$

**Lemma A.4.** *Consider one execution of* APPROX-DD $(G, \epsilon)$ *in Algorithm 1. For every $p \in (0,1)$, with probability at least $1 - p$ the subroutine* SHAVE *is called* $O\left(\frac{\log\left(\left(\epsilon+\frac{1}{\epsilon}\right)\log\frac{n}{p}\right)\log\frac{n}{p}}{\log(1+\epsilon)}\right)$ *times.*

*Proof.* Let $T = 24(1+\alpha)(1+\beta)\ln\frac{n}{p}$ and consider the number of calls to PEEL such that $\Delta \ge T$. Since $24\ln\frac{n}{p} = 12\ln\frac{n^2}{p^2} > 12\ln\frac{n}{2p}$ for all $n \ge 2$, by Lemma 3.2 each such call with probability at least $1 - \frac{p}{2n}$ invokes SHAVE at most once. Moreover, by Lemma A.3 the number of those calls is at most

$$\min\left(n, \frac{\log n}{\log(1+\alpha)}\right) = \min\left(n, \frac{\log n}{\log(1+\epsilon/2)}\right) \tag{12}$$

By a union bound over the at most $n$ calls, thus, with probability at least $1 - \frac{p}{2}$ the overall number of calls to SHAVE performed while $\Delta \ge T$ is at most $\frac{\log n}{\log(1+\epsilon/2)}$.

Now suppose $\Delta < T$. By Lemma A.3 PEEL is called $O\left(\frac{\log T}{\log(1+\epsilon/2)}\right)$ times before $G = \emptyset$, and by Lemma 3.2 each such call invokes SHAVE $O(\log\frac{n}{p})$ times with probability at least $1 - \frac{p}{n}$. Again by the union bound, with probability $1 - \frac{p}{2}$ the overall number of calls to SHAVE for $\Delta < T$ is in

$$O\left(\frac{\log T}{\log(1+\epsilon)} \cdot \log\frac{n}{p}\right) = O\left(\frac{\log\left(\left(\epsilon+\frac{1}{\epsilon}\right)\log\frac{n}{p}\right)\log\frac{n}{p}}{\log(1+\epsilon)}\right) \tag{13}$$

where we used $T = O\left(\left(\epsilon + \frac{1}{\epsilon}\right)\log\frac{n}{p}\right)$. By one final union bound over the two cases $\Delta \ge T$ and $\Delta < T$, with probability $1 - p$ the expression in (13) bounds the total number of calls to SHAVE. $\quad\square$

## A.1  Proof of Theorem 3.1

The number of passes is dominated by the number of calls to the subroutine COMPUTEALLDEG. Indeed, since $V = \{1, \ldots, n\}$, we can keep a global array of $O(n)$ words that for every $v \in V$ stores the following information: whether $v$ is still in $G$, the index $i$ of the subset $H_i$ or $R_i$ or $S_i$

containing $v$, as well as the degree of $v$ in any of those subgraphs. After each pass, the whole array can be updated in linear time so to be ready for the next iteration. It is then easy to see that COMPUTEALLDEG can be ran on any graph $G$, $H_i$, etc. using one pass and linear time: scan the edge list, and for each edge $e = \{u, v\}$ increase the degree counters of both $u$ and $v$ *iff* they both belong to the target vertex set (which can be checked in time $O(1)$ using the array). This also shows that the total running time per pass of APPROX-DD is in $O(m)$.

It remains to bound the total number of calls to COMPUTEALLDEG. First, assume $M = \Omega(n)$. In this case we simply run APPROX-DD until it terminates when $V = \emptyset$. By Lemma A.4, using $p = n^{-\Theta(1)}$, the total number of calls fo SHAVE is in

$$O\left(\frac{\log\left(\left(\epsilon + \frac{1}{\epsilon}\right)\log n\right)\log n}{\log(1 + \epsilon)}\right) = O\left(\frac{\log n}{\epsilon}\log\frac{\log n}{\epsilon}\right) \tag{14}$$

As by Lemma A.2 each call to SHAVE makes $O(1/\epsilon)$ invocation of COMPUTEALLDEG, the bound follows.

Now assume $M = \Omega(n\log n)$. Let again $p = n^{-\Theta(1)}$. In this case we let APPROX-DD stop as soon as $\Delta < T$, where

$$T = 12(1 + \alpha)(1 + \beta)\ln\frac{n}{p} = O\left(\frac{\log n}{\epsilon}\right) \tag{15}$$

as defined in Lemma 3.2. At that point we use one additional pass to store $G$, and use the linear-time bucketing algorithm of [Bre23]. By Lemma 3.2, w.h.p. every call of PEEL makes only one call to SHAVE. Moreover, by Lemma A.3 APPROX-DD makes $O(\log_{1+\epsilon} n) = O\left(\frac{\log n}{\epsilon}\right)$ calls to PEEL. Hence the total number of calls to SHAVE is in $O\left(\frac{\log n}{\epsilon}\right)$. As by Lemma A.2 each call to SHAVE makes $O(1/\epsilon)$ invocation of COMPUTEALLDEG, the bound follows.

### A.2 Pseudocode of PREPROCESS

---

**Algorithm 4** PREPROCESS

---

1: **function** PREPROCESS( )
2:      compute ordering $\prec \leftarrow$ APPROX-DD$(G, \epsilon)$          ▷ see Algorithm 1
3:      $(\vec{p}, \Gamma) \leftarrow$ COMPUTEDISTRIB$(\prec)$          ▷ see Algorithm 2
4:      **return** $(\prec, \vec{p}, \Gamma)$

---

## B Proofs for Section 3.2

**Lemma 3.3.** *In the semi-streaming model with $M = \Omega(kn)$, the routine* COMPUTEDISTRIB *(Algorithm 2) can be implemented so as to use 1 passes and time $O(m + n \cdot k^2 \log k)$.*

*Proof.* Computing $Z$, $\vec{p}$, and $\Gamma$ clearly takes time $O(n)$, hence we focus on computing $d(v|G(v))$ and $\mathbb{1}_v$ for all $v \in V$. Consider the oriented graph $\vec{G} = (V, A)$ constructed from $G$ as follows. For every $v \in V$ say that $v$ is *light* if $d(v|G(v)) < k - 1$ and *heavy* otherwise. For every light vertex $v$, add the arc $(v, x)$ to $A$ for every $x \in \mathcal{N}(v|G(v))$ (let us call such arc a *light arc*). Moreover, if $v$ is light and has heavy neighbors in $G$, then add to $A$ the arc $(v, y)$ where $y$ is largest heavy neighbor of $v$ in $\prec$ (let us call such arc a *heavy arc*). For every $v$ assign a weight $w(v) = 1$ if $v$ is light and $w(v) = k$ if $v$ is heavy. Moreover, for every $v$ let $\vec{G}(v)$ be the subset of vertices reachable from $v$ in $G(v)$ by following the arcs in $\vec{G}$. Finally, define $\vec{w}(v) = \sum_{u \in \vec{G}(v)} w(u)$.

We claim that $\vec{w}(v) \geq k$ if and only if $|\mathcal{G}(v)| \neq \emptyset$. To see this, note that both conditions are equivalent to the following one: in $\vec{G}$, starting from $v$ and following forward arcs to vertices in $G(v)$, one can reach at least $k$ light vertices or at least one heavy vertex. Thus we need only to construct $\vec{G}$ and, from each $v \in V$, run a truncated BFS to check if $\vec{w}(v) \geq k$. Assuming $\vec{G}$ is represented by adjacency

lists, each BFS requires $O(k^2 \log k)$ operations, since it stops before seeing $k^2$ arcs. The total time for the BFS is thus $n \cdot k^2 \log k$.

To construct $G$, we use a single pass to compute $d(v|G(v))$ for each vertex $v$ as well as determine heavy vertices and light vertices. We initially mark all vertices as light vertices. As we process the edges, for each edge $(v, x)$ from $v$ to $x \in \mathcal{N}(v|G(v))$ where $v$ is light, we add it to $A$ as a light arc. If $v$ is heavy, we update the heavy arc of $x$ if $x$ is light. Once we detect any light vertex $v$ having at least $k - 1$ light arcs, we mark $v$ as a heavy vertex, update the heavy arc of the other endpoint of each of these light arcs and remove them. This can clearly be implemented in time $O(m)$. $\qquad\square$

## C   Proofs for Section 3.3

### C.1   Proof of Theorem 3.4

Each instance of SAMPLE requires only $O(k^2)$ words of memory. Hence, with memory $M$, we can run $Q = \Theta(M/k^2)$ parallel instances of SAMPLE. By Lemma C.1, each instance independently returns a uniform random $k$-graphlet with probability at least $\frac{k^{-O(k)}}{(1+\epsilon)^{k-1}} = k^{-O(k)}$. By standard concentration bounds, then, with high probability the number of uniform random $k$-graphlets returned overall is $\Theta(M/k^{O(k)})$.

We prove the two bounds on the number of passes and running time by describing two implementations of SAMPLE. Both implementations use the same approach for PROB, which is ran in parallel for all $Q$ instances and takes only one pass. First, PROB computes $G[S]$ (i.e., it retrieves the edges) as well as $d(u|G(v))$ for all $u \in S$. To this end each instance creates pairs $(u, v), (v, u)$ for each $\{u, v\} \in \binom{S}{2}$ and appends them to a global list $L$, where $|L| \le O(Qk^2) = O(M)$. Then, we invoke CHECKEDGES$(L)$; afterwards, every instance finds each one of its edges in the sorted output list. The same is for SUBDEGREES. By Lemma C.2 and Lemma C.3 we can carry out all of this in one pass and total time $O((|L| + m) \log |L|)$. Since $|L| \le M = O(m)$, the time to compute $G[S]$ and $d(u|G(v))$ for all $u \in S$ over all $Q$ instances is in $O(m \log n)$.

Each instance of PROB can then conclude its execution by running the loop of line 18. This loop iterates over all subsets of $S$, which are at most $2^k$, and performs $O(k^2)$ operations for each of them. Summed over the $Q = O(M/k^2)$ instances, this yields a time in $O(2^k M)$. Adding the bound above for computing $G[S]$ and the degrees, we obtain a total running time of $O(M2^k + m \log n)$ for all instances of PROB.

It remains to discuss the implementations of RAND-GROW.

*First implementation.* This is the straightforward approach in which the $Q$ instances do not share any computation. The running time of RAND-GROW is dominated by the iteration of the loop at line 9. Every edge in the stream is sent to each of $Q$ instances. The instance selects only the edges that belong to $E_i$, which requires time $O(k)$ per edge; and it uses reservoir sampling to sample from those edges uniformly at random. Hence, over the $Q$ instances, the time for each pass is $O(Qkm)$, and the time for making the $k - 1$ passes is $O(Qk^2m) = O(Mm)$.

Since we consider $k = o(\log n)$ and $M = \Omega(nk)$, the time for PROB is $O(M2^k + m \log n) = O(Mm)$. Hence, the total running time to execute the batch is $O(Mm)$.

*Second implementation.* For a given instance of RAND-GROW $(v, \prec)$ consider a generic iteration of the loop at line 9. As in the original implementation of [Bre23], we first sample a node $u \in S_i$ with probability proportional to the number of edges it has in $E_i$, and then sample one of its edges $\{u, u'\}$ in $E_i$ uniformly at random. to sample $u$, first we compute $d(u|G(v))$ for every $u \in S$, as well as $G[S]$. By using again SUBDEGREES and CHECKEDGES, this requires one pass and time $O(m \log n)$. Then, we can sample $u$ with probability proportional to $d(u|G(v)) - d(u|G[S])$. This takes time $O(k^2)$, as computing $d(u|G[S])$ for every $u$ entails checking the edges of $G[S]$. Next, to draw an edge $\{u, v'\}$ uniformly at random in $E_i$, we draw a random uniform subset of $i$ edges $e_1, \ldots, e_i$ incident with $u$ in $E_i$. Since at most $i - 1$ of them can be incident with $S_i$, it follows that at least one of them is not, and is therefore a random uniform edge incident with $u$ in the cut of $E_i$.

To draw $e_1, \ldots, e_i$ we use SAMPLEEDGES (Algorithm 7).  First, we draw a random uniform subset of $i$ distinct integers $j_1, \ldots, j_i$ from $\{1, \ldots, d(u|G(v))\}$. We then create $i$ triplets

$(u, v, j_1), \ldots, (u, v, j_i)$. For each $h = 1, \ldots, i$ we then create a global list $T_h$ consisting of the triplets $(u, v, j_h)$. Finally, we execute SAMPLEEDGES$(T_h)$ for all $h = 1, \ldots, i$ in parallel. Let $F_h$ be the dictionary returned by SAMPLEEDGES$(T_h)$. By Lemma C.4, and by the choices of $j_1, \ldots, j_i$, each $e_h = F_h[(u, v)]$ is a random uniform edge between $u$ and $\mathcal{N}(u|G(v))$, for $h = 1, \ldots, i$. Moreover, and crucially, Lemma C.4 guarantees that $e_1, \ldots, e_i$ are all distinct, since the instances of SAMPLEEDGES receive the same pairs $(x, y)$; only the indices $j$ differ. Thus $e_1, \ldots, e_i$ is a random uniform subset of $i$ edges incident with $u$ in $E_i$, as desired. Again by Lemma C.4, as $\sum_{h=1}^{i} |T_h|$ is again bounded by $M = O(m)$, the total time is in $O(m \log n)$.

Therefore we have 2 passes for each one of the $k - 1$ iterations of RAND-GROW, plus one pass for PROB, for a total of $2k - 1$ passes. The total running time is $O(M2^k + km \log n)$.

## C.2 Obtaining Degrees and Edges, in Parallel

In this subsection, we elaborate the details for subroutines that are used in the proof of Theorem 3.4. First, we prove that SAMPLE returns independent uniform graphlets, and with a probability that is in $(k(1 + \epsilon))^{-O(k)}$.

**Lemma C.1.** *After running* PREPROCESS *once, every invocation of* SAMPLE *returns a size-$k$ set $S$ with probability at least $\frac{1}{(1+\epsilon)^{k-1}((k-1)!)^4}$. Moreover, if* SAMPLE *succeeds, then the graphlet returned is drawn uniformly at random from the set of all $k$-graphlets of $G$.*

*Proof.* We adapt the claims and proofs of Section 6 of [Bre23] as follows. Suppose the vertices of $G$ is sorted according to a $\frac{1}{1+\epsilon}$-DD ordering $\prec$, and suppose the subset $S$ forms a $k$-graphlet of $G$, where $v \in S$ is the earliest vertex among $S$ in the ordering. First, one can verify that [Bre23, Lemma 24] can be generalized to yield the result that for $v \in S_i \subseteq S$ such that $i = |S_i|$,

$$\frac{d(v|G(v))}{i} \leq |\operatorname{Cut}(S_i, G(v) \setminus S_i)| \leq i(1 + \epsilon)d(v|G(v)). \tag{16}$$

Given $v$ and $\prec$, suppose $q_S$ is the probability that RAND-GROW$(v, \prec)$ returns $S$. Using the inequality in (16), one can verify that [Bre23, Lemma 25] can be generalized to yield

$$\frac{1}{(k-1)!} \left( \frac{1}{(1+\epsilon)d(v|G(v))} \right)^{k-1} \leq q_S \leq (k-1)!^3 \left( \frac{1}{d(v|G(v))} \right)^{k-1}. \tag{17}$$

One can also check that PROB$(v, S, \prec)$ computes precisely $q_S$. Now consider the probability that $S$ is accepted in line 5 of Algorithm 3, $p_{acc}(S) = \frac{\Gamma}{p(v) \, q_S}$. The same arguments of [Bre23] show that

$$p_{acc}(S) \geq \frac{1}{(1 + \epsilon)^{k-1}((k - 1)!)^4}. \tag{18}$$

Moreover, the probability that $S$ is sampled *and* returned by SAMPLE is $p(v) \cdot q_S \cdot p_{acc}(S) = \Gamma$, which is independent of $S$. This implies that given that SAMPLE successfully returns some subset $S$, $S$ is drawn uniformly at random from the set of all $k$-graphlets of $G$.

$\square$

Next, we give a simple routine for checking efficiently for a set of edges, in parallel. This routine, and other ones below, operate on list of pairs of vertices $(u, v)$. When we speak of an order over those pairs, we mean the lexicographic order induced by the approximate DD order $\prec$: $(u, v) \prec (u', v')$ iff $u \prec u'$, or $u = u'$ and $v \prec v'$.

**Lemma C.2.** *Let $L = (u_1, v_1), \ldots, (u_t, v_t)$ where $u, v \in V$ for each $(u, v) \in L$. Then,* CHECKEDGES$(L)$ *makes one pass, runs in time $O((|L| + m) \ln |L|)$, and returns the list all edges of $G$ that are in $L$.*

*Proof.* The sorting of $L$ takes $O(|L| \log |L|)$ time. For each of the $m$ edges in the list, searching in the sorted list takes $O(\log |L|)$ time. Finally, removing unmarked edges takes $O(|L|)$ time. $\square$

---

**Algorithm 5** CHECKEDGES

---

1: **function** CHECKEDGES($L = (u_1, v_1), \ldots, (u_t, v_t)$)
2:     Sort $L$ and remove duplicates.
3:     **for** each orientation $(u, v)$ of every edge in the stream **do**
4:         Search $(u, v) \in L$ and, if present, mark it.
5:     Delete all unmarked elements from $L$.
6:     **return** $L$

---

We give two routines, SUBDEGREES and SAMPLEEDGES, that respectively compute a set of degrees $d(u|G(v))$ and sample a set of uniform random edges from subgraphs $G(v)$. The routines make use of sorted lists $L$ of tuples, where in each tuple the first two elements are vertices $u, v$. The *segment* $L(u, v)$ is the sublist formed by all elements whose first two entries are $u, v$, in this order, and $L(u)$ is the sublist formed by all sublists $L(u, v)$. The maps returned by the algorithms are assumed to support lookup in time $O(\log |L|)$.

---

**Algorithm 6** SUBDEGREES

---

1: **function** SUBDEGREES($\prec, L = (u_1, v_1), \ldots, (u_t, v_t)$)
2:     Sort $L$ by $\prec$ and remove duplicates.
3:     Initialize $c_{u,v} \leftarrow 0$ for every $(u, v) \in L$.
4:     **for** each orientation $(u, \hat{v})$ of every edge in the stream **do**
5:         **if** $\exists (u, v) \in L, (u, v) \preceq (u, \hat{v})$ **then**
6:             Let $(u, v)$ be the largest such pair.
7:             Update $c_{u,v} \leftarrow c_{u,v} + 1$.
8:     Init dictionary $D : V^2 \rightarrow [n]$.
9:     **for** every segment $L(u)$ and every $(u, v) \in L(u)$ **do**
10:        Set $D(u, v) \leftarrow \sum \{c_{u,w} : (u, w) \in L(u) \wedge (u, w) \succeq (u, v)\}$
11:     **return** dictionary $D$

---

**Lemma C.3.** *Let* $L = (u_1, v_1), \ldots, (u_t, v_t)$ *where* $u, v \in V$ *for each* $(u, v) \in L$. *Then* SUBDEGREES$(\prec, L)$ *makes one pass, runs in time* $O((|L| + m) \log |L|)$, *and returns a dictionary* $D$ *such that* $D(u, v) = d(u|G(v))$ *for all* $(u, v) \in L$.

*Proof.* The algorithm makes one pass by construction. For the running time, note that the loop of line 9 requires time $O(|L|)$ by scanning each $L(u)$ in reverse order. It is easy to see that the rest of the algorithm requires time $O((|L| + m) \log |L|)$. Now consider any pair $(u, v) \in L$. Every edge in the stream between $u$ and a vertex $w$ in $G(v)$ is considered as $(u, w)$ by the loop of line 4. It then contributes 1 to precisely one counter $c_{u,w'}$ for some $w' \succeq v$. Therefore $(u, w)$ also contributes 1 to $D(u, v)$, by the loop of line 9. It follows that $D(u, v) = d(u|G(v))$. $\square$

**Lemma C.4.** *Let* $T = \{(u_1, v_1, j_1), \ldots, (u_t, v_t, j_t)\}$ *be such that, for every* $(u, v, j) \in T$, *we have* $u, v \in V$ *with* $v \preceq u$, *and* $j \in \{1, \ldots, d(u|G(v))\}$. *Then,* SAMPLEEDGES$(\prec, T)$ *makes one pass over the edges of* $G$, *and runs in time* $O((|T| + m) \log |T|)$.

*Moreover, for every* $(u, v, j) \in T$, *the map* $F(u, v, j)$ *returns the* $j$-th *edge incident to* $u$ *in* $G(v)$, *where the ordering of* $u$'s *incident edges depends only on (i) the pairs* $(u_i, v_i)$ *appearing in* $T$ *(i.e., the first two components of each triple in* $T$) *and (ii) the order of the edges in the stream.*

*Proof.* The algorithm makes one pass by construction. For the running time, it can be checked that several steps involve sorting a data structure of size $O(|T|)$, and hence, they take $O(|T| \log |T|)$ time. For the loop at line 20, there are $O(m)$ iterations, where each iteration involves search an indexed data structure of size $O(|T|)$, which takes $O(\log |T|)$ time. Hence, the total time is $O((|T| + m) \log |T|)$.

We next prove correctness. For every $(u, v) \in L$, let $E(u, G(v))$ be the set of incident edges of $u$ in $G(v)$. We next specify an ordering on $E(u, G(v))$ that depends only on the (other) pairs in $L$ and the order of the edges in the stream. Let $L_{\succeq}(u) = \{(u, v_1), \ldots, (u, v_h)\}$ be the sorted list of pairs $(u, v_i)$ in $L$ such that $v_i \succeq v$, where $v = v_1$.

---
**Algorithm 7** SAMPLEEDGES
---
1: **function** SAMPLEEDGES($\prec$, $T = (u_1, v_1, j_1), \ldots, (u_t, v_t, j_t)$)
2:      Compute the list $L$ of distinct pairs $(u, v)$ s.t. $\exists (u, v, j) \in T$, and sort $L$ by $\prec$
3:      **for** each $(u, v) \in L$ **do**
4:          Compute the increasingly sorted list

$$J(u, v) := \{j : (u, v, j) \in T\}.$$

5:      Use SUBDEGREES($\prec$, $L$) to compute $d(u|G(v))$, $\forall (u, v) \in L$.

6:      Init empty list $L^*$.
7:      Init dictionary $J^*$ mapping each $(u, v) \in L$ to an empty list.
8:      Init dictionary $D^* : V^2 \times [n] \to V^2 \times [n]$
9:      **for** every $(u, v) \in L$ and every $j \in J(u, v)$ **do**
10:          Find the largest $(u, v')$ in $L(u, v)$ such that

$$d(u|G(v')) - d(u|G(v)) + j \geq 1.$$

11:          $j' \leftarrow d(u|G(v')) - d(u|G(v)) + j$
12:          Append $(u, v')$ to $L^*$
13:          Append $j'$ to $J^*(u, v')$
14:          $D^*(u, v, j) \leftarrow (u, v', j')$
15:      Sort $L^*$ by $\prec$ and remove duplicaes
16:      For each $(u, v) \in L^*$, sort $J^*(u, v)$ in increasing order and remove duplicates

17:      Init dictionary $D : V^2 \times [n] \to E(G)$
18:      **for** each $(u, v) \in L^*$ **do**
19:          init a counter $c_{u,v} = 0$
20:      **for** each orientation $(u, \hat{v})$ of every edge in the stream **do**
21:          **if** $\exists (u, v) \in L^*$ with $(u, v) \preceq (u, \hat{v})$ **then**
22:              Let $(u, v)$ be the largest such pair
23:              $c_{u,v} \leftarrow c_{u,v} + 1$
24:              **if** the first element $j$ of $J^*(u, v)$ is $j = c_{u,v}$ **then**
25:                  Set $D(u, v, j) \leftarrow \{u, \hat{v}\}$
26:                  Delete $j$ from the head of $J^*(u, v)$
         Define dictionary $F : V^2 \times [n] \to E(G)$ s.t. $F(u, v, j) := D(D^*(u, v, j))$.
27:      **return** dictionary $F$
---

For each $i = 1, \ldots, h$, let $E^i(u, G(v))$ be the subset of $E(u, G(v))$ formed by all edges $(u, w)$ where $v_i \preceq w \prec v_{i+1}$ (for $i = h$ just consider $v_i \preceq w$). Clearly, the sets $E^i(u, G(v))$ form a partition of $E(u, G(v))$; in fact,

$$E^i(u, G(v)) = E(u, G(v_i)) \setminus E(u, G(v_{i+1}))$$

with again the exception $E^h(u, G(v)) = E(u, G(v_h))$.

The order on $E(u, G(v)$ is given as follows. Start with the elements of $E^1(u, G(v))$ in the order they appear in the stream. Then, add the elements of $E^2(u, G(v))$ in the order they appear in the stream, and so on until $E^h(u, G(v))$. Note that this order depends only on the pairs $(u, v) \in L$, and on the order of the edges in the stream.

Note that if $j$ is uniform at random in $\{1, \ldots, d(u|G(v))\}$, and independent from both $L$ and the stream, then the $j$-th edge $e_j$ in that order is uniform at random in $E(u, G(v))$.

Now let us look at the set $E^{i_j}(u, G(v))$ that contains $e_j$. By standard computations,

$$i_j = \max\{i : d(u|G(v_i)) - d(u|G(v)) + j \geq 1\}$$

Note that $i_j$ depends only on $j$, and on the sizes of the sets $E^i(u, G(v))$, and not on the order of the elements in the stream. The vertex $v_{i_j}$ is precisely the vertex $v'$ computed by the algorithm at line 9. Moreover, the index $j'$ computed at the same line is precisely:

$$j' = j - d(u|G(v)) + d(u|G(v_j))$$

It follows that, if we map $(u, v)$ to the $j'$-th edge of $E^{i_j}(u, G(v))$ that appears in the stream, which is $e_j$, then the algorithm is correct. This is precisely what the algorithm does in the loop of line 20. The counter $c_{uv}$ keeps track, for every $i = 1, \ldots, h$ as above, of the number of edges of $E^i(u, G(v))$ seen so far; when $c_{uv}$ matches some $j$, it maps $(u, v)$ to $e_j$, as desired. □

## D    Proofs for Section 4

**Lemma 4.1.** *Given a graph $G = (V, E)$, $\alpha \in (0, 1)$, any $q$-pass streaming algorithm computing a node $v$ such that $\delta_G(v) \geq \alpha \max_{w \in V} \delta_G(w)$ with probability at least $\frac{2}{3}$, requires memory $\Omega\left(\frac{\alpha^2 |V|}{q}\right)$.*

*Proof.* Let $\mathcal{A}$ be a $q$-pass $s$-space streaming algorithm for computing a vertex with degree being at least a fraction of $\alpha$ of the maximum degree. Without loss of generality, we assume that $\mathcal{A}$ computes its degree as well (as one additional pass and $O(\log n)$ memory suffices to compute the degree of a given node). We will use $\mathcal{A}$ to obtain a multi-agent protocol for the disjointness problem.

Let $t = \lceil \frac{1}{\alpha} \rceil + 1$ be the number of agents in a one-way multi-party communication protocol. Recall that in the disjointness problem, we may assume that either the supports of the bit vectors are pairwise disjoint (NO instance) or they share one common element but are otherwise disjoint (YES instance). From the $n$-bit vectors from the $t$ agents, we obtain a data stream of a bipartite graph $G = (U, V, E)$ as follows.

- $V = \{v_1, \ldots, v_n\}$ corresponding to the $n$ bits

- For each agent $i$, we introduce a set of vertices and edges associated with $i$ as follows. For every $x_i$, for every $j \in [n]$ s.t. $x_{i_j} = 1$, there is a node $u_{i_j} \in U$ as well as an edge $(u_{i_j}, v_j) \in E$.

- The edges are arranged in the stream so that the edges associated with a same agent $i$ appear next to each other in the stream while preceding the edges associated with agent $i + 1$. In other words, $(u_{i_j}, v_j)$ precedes $(u_{\ell_h}, v_h)$ in the stream if $i < \ell$, with all other edges being arranged in an arbitrary order.

The protocol for the disjointness problem simulates $\mathcal{A}$ as follows. The first player executes $\mathcal{A}$ on the set of edges associated with her, i.e. $\{(u_{1_j}, v_j) : j \in [n], (u_{1_j}, v_j) \in E\}$. Then, she sends her memory snapshot (with size $O(s)$) to the second player that continues the execution of $\mathcal{A}$ on the set of edges $\{(u_{2_j}, v_j) : j \in [n], (u_{2_j}, v_j) \in E\}$. The second player in turn sends her memory snapshot to the third player and so forth until reaching the $t$-th player, which sends her memory snapshot back to the the 1st player. This procedure is iterated $q$ times at the end of which the $t$-th player outputs "YES" if the degree of the node computed by $\mathcal{A}$ is $\geq 2$, and "NO" otherwise.

Such a protocol is a correct protocol for the disjointness problem, in that, by construction, the maximum degree of any node in $G$ is either $t$ ( in the YES instance ) or 1 ( in the NO instance ). Hence, with probability at least $\frac{2}{3}$, in a YES instance $\mathcal{A}$ finds a node with degree at least $\alpha \cdot t > 1$, while in a NO instance such a node has degree 1.

Observe that the number of vertices in the bipartite graph is at least $n = |V|$ while it holds that $|U| + |V| = O(t + n) = O(n)$. Moreover, the communication cost $|\Pi|$ of the protocol satisfies $|\Pi| = O(s \cdot t \cdot q) = O\left(\frac{s \cdot q}{\alpha}\right)$ by our arguments, and $|\Pi| = \Omega\left(\frac{n}{t}\right) = \Omega(\alpha n)$ by the bounds above. From this, we derive $s \in \Omega\left(\frac{\alpha^2 n}{q}\right)$.

□

**Lemma 4.2.** *Given a graph $G = (V, E)$, $k \geq 3$, any $q$-pass streaming algorithm for the uniform $k$-graphlet sampling problem on $G$ requires memory $\Omega\left(\frac{|V|}{q}\right)$.*

*Proof.* We use the same construction of Lemma 4.1 while letting $t = k - 1$. In such a case, a "NO" instance for the disjointness problem corresponds to a graph with maximum degree one (with each connected component consisting of either a singleton or a single edge), while a "YES" instance

corresponds to a graph containing a connected component being a *star* with $t + 1 = k$ nodes, with all other connected components having maximum degree one. Observe there is exactly one $k$-graphlet in the "YES" instance, with no $k$-graphlet in the "NO" instance. As a result, any streaming algorithm succeeding in sampling uniformly a $k$-graphlet with probability $\delta$ would also solve the disjointness problem. $\qquad\square$

# E  The Heuristic Approx-DD

In this section we introduce a heuristic approach to substitute Approx-DD, which efficiently reduces the number of passes while preserving the probabilistic upper bounds in Theorem 3.1.

The heuristic approach aims to extract a large subgraph of high-degree vertices into memory, so that it can run a simple greedy algorithm that removes the vertex with the largest degree in each iteration without requiring any additional passes. The process starts from finding this subgraph and determining whether running the heuristic is worthwhile. First, we sort vertices in descending order of $d(v|G)$. Let the sorted vertex list be $v_1, \cdots, v_{|V(G)|}$. We use a streaming pass to compute $d(v_i|\bar{G}(v_i))$ for each vertex $v_i$, where $\bar{G}(v_i)$ represents the induced subgraph of the prefix $v_1, \cdots, v_i$. We know that $\sum_{j=1}^{i} d(v_j|\bar{G}(v_j))$ is the number of edges in $\bar{G}(v_i)$. Let $i^*$ be the largest $i$ such that edges in $\bar{G}(v_i)$ can be stored in memory, i.e., $\sum_{j=1}^{i} d(v_j|\bar{G}(v_j)) \leq M$. If $i^* = |V(G)|$ or $d(v_{i^*+1}|G) \leq \frac{\Delta}{1+\alpha}$, we run the heuristic algorithm Peel-Heuristic instead, otherwise we still run Peel.

In Peel-Heuristic, we first use a streaming pass to record all edges in $\bar{G}(v_{i^*})$. Then we repeatedly select the vertex $v$ in $\bar{G}(v_{i^*})$ with the maximum $d(v|G)$, append it to the DD order, remove it from $\bar{G}(v_{i^*})$ and update $d(v'|G)$ for all vertices $v' \in G(v_{i^*})$ that are connected to $v$. We continue this process until $d(v|G)$ of every remaining vertex $v$ in $\bar{G}(v_{i^*})$ are less than $\frac{d(v_{i^*+1}|G)}{1+\epsilon}$.

The idea is summarized in Algorithm 8.

It is easy to see that whenever we append $v$ to $L$, it holds that $d(v|G(v)) = d(v|G) \geq d(u|G) = d(u|G(v))$ for each $u \in \bar{G}(v_{i^*})$, and $d(v|G(v)) \geq \frac{d(v_{i^*+1}|G)}{1+\epsilon} \geq \frac{d(v_j|G(v))}{1+\epsilon}$ for each $v_j$ where $j > i^*$, therefore the final order we obtain is indeed a $\frac{1}{1+\epsilon}$-DD order. Note that after an execution of Peel-Heuristic, the degrees of all remaining vertices are no greater than $d(v_{i^*+1}|G) \leq \frac{\Delta}{1+\alpha}$. This means that Peel-Heuristic performs no worse than Peel if the condition in line 9 is satisfied, and Theorem 3.1 still bounds the number of passes for Approx-DD-Heuristic. However, since Peel-Heuristic only requires 3 passes on average, and it removes more vertices and reduces the maximum degree more effectively than Peel in sparse graphs, Approx-DD-Heuristic performs much better than Approx-DD in practice.

---
**Algorithm 8** Approximate DD order heuristically. The lines marked with a $\triangleright$ entail a streaming pass.
---
1: **function** APPROX-DD-HEURISTIC($G, \epsilon$)
2:   $L \leftarrow$ empty list
3:   $\alpha \leftarrow \frac{\epsilon}{2}$
4:   **while** $V(G) \neq \emptyset$ **do**
5:    compute $d(v|G)$ for all $v \in V(G)$               $\triangleright$
6:    $v_1, \cdots, v_{|V(G)|} \leftarrow$ the sorted list of $V(G)$ in descending order of $d(v|G)$
7:    compute $d(v_i|\bar{G}(v_i))$ for each $i \in [|V(G)|]$          $\triangleright$
8:    $i^* \leftarrow \max\{i| \sum_{j=1}^{i} d(v_j|\bar{G}(v_j)) \leq M\}$
9:    **if** $i^* = |V(G)| \vee d(v_{i^*+1}|G) \leq \frac{\Delta}{1+\alpha}$ **then**
10:     $b \leftarrow \frac{d(v_{i^*+1}|G)}{1+\epsilon}$ if $i^* < |V(G)|$ otherwise $0$
11:     $(G, L) \leftarrow$ PEEL-HEURISTIC($G, \{v_1, \cdots, v_{i^*}\}, b, \epsilon, L$)
12:    **else**
13:     $(G, L) \leftarrow$ PEEL($G, \epsilon, L$)
14:   **return** $L$

15: **function** PEEL-HEURISTIC($G, P, b, \epsilon, L$)
16:   $E(P) \leftarrow$ the edges whose both endpoints are in $P$         $\triangleright$
17:   **while** $P \neq \emptyset$ **do**
18:    $v \leftarrow \arg\max_{v \in P}\{d(v|G)\}$
19:    **if** $d(v|G) < b$ **then**
20:     **break**
21:    append $v$ to $L$
22:    $G \leftarrow G \setminus \{v\}, P \leftarrow P \setminus \{v\}$
23:    **for** each $v' \in P$ such that $(v, v') \in E(P)$ **do**
24:     $d(v'|G) \leftarrow d(v'|G) - 1$
25:   **return** $(G, L)$
---

