# OpenReview forum: "Efficient Streaming Algorithms for Graphlet Sampling"
_NeurIPS.cc/2024/Conference — NeurIPS 2024 poster_

### Official Review · Reviewer_5Ka2 · 2024-07-06

**Soundness:** 4
**Presentation:** 4
**Contribution:** 2
**Rating:** 5
**Confidence:** 4

**Summary:**

The paper presents algorithms for sampling a connected subgraph on k nodes, the so called k-graphlets, from a massive graph revealed as a stream of edges in arbitrary order. The algorithms work in the semi-streaming model of computation where one can only use linear space in the number of nodes. The algorithms are allowed several passes over the graph and sample k-graphlets uniformly at random. The computational complexity of the approach and the sampling guarantees are formally analyzed and several theorems show that the approach has a nearly optimal computational complexity.

**Strengths:**

- A strong algorithmic contribution.
- Solid and apparently sound theoretical analysis. In particular, lower bounds on the computational complexity that almost match the complexity of the proposed approach.
- A very well-written paper.

Note: The overall low score I give for the contribution is based on the question "Are the results valuable to share with the broader NeurIPS community?" Otherwise, the score is "Excellent".

**Weaknesses:**

Overall, I am not convinced that NeurIPS is the right venue for a graph mining paper. The weaknesses listed below are in the context of this context.

- There is no attempt to connect the results to machine learning applications. Clearly, from an algorithmic point of view, uniform sampling of k-graphlets is an important problem. But does it yield anything useful for downstream applications? Maybe the k-graphlet distribution is dominated by k-paths and one would need many samples in order to estimate the distribution of the different kinds of k-graphlets that is necessary for graph classification algorithms to yield good results.
- I understand this is the first algorithm for the problem but I still believe it can be compared to some heuristics in order to show the advantages of uniform sampling. Off the top of my head, I can think of the monochromatic sampling approach by Pagh and Tsourakakis. For a sparsified graph, count the number of k-graphlets. The approach will not guarantee uniform sampling but might still yield useful estimates and might be more efficient.

**Questions:**

Comment on the above weaknesses. How would you convince a machine learning researcher to read your paper?

**Limitations:**

The limitations are correctly addressed from an algorithmic point of view. But no attempt is made to consider the limitations when applying the approach in a machine learning setting, for example graph classification.

---

> ### Author Rebuttal · Authors · 2024-08-07
>
> We would like to thank the reviewer for the comments, which we would like to address as follows.
>
> [Weakness 1 and Limitation]
>
> In the next version of our work, we will highlight the connection between machine learning and the graph mining problem in our paper as follows:
>  - Graphlet sampling can be used to select representative subgraphs for training, validating and testing (e.g. in a neural network), which can speed up the training process and reduce computational requirements significantly.
>  - Graphlet sampling can be used to extract meaningful features from large graphs. The frequency distribution of different graphlets can be used as a feature vector to represent the graph structure [MSOI+02], which can be approximately obtained by our approach.
>
> Other applications can be also found in the literature such as graph classification [TLT+19], graph kernels [SVP+09], and graph neural networks [PLG+20].
>
> [Weakness 2]
>
> Thank you for the suggestion of using a sparsified graph. However, for a k-graphlet that has only k-1 edges in the induced subgraph, the sparsification process will likely make this graphlet disconnected.  Even though we do not immediately see how this idea will work, we think it is an interesting future research direction.

---

> > ### Comment · Reviewer_5Ka2 · 2024-08-09
> > **Response to author's rebuttal**
> >
> > I thank the authors for their response. Again, this is a strong paper and if the authors indeed put some effort into bridging the gap between algorithmic data mining and machine learning, I think it should be accepted. I am raising my score, I guess it is now in the hands of the area chair to decide if the paper would be interesting to the NeurIPS community.

---

### Official Review · Reviewer_T1eM · 2024-07-09

**Soundness:** 3
**Presentation:** 3
**Contribution:** 2
**Rating:** 5
**Confidence:** 3

**Summary:**

Given a graph $G$, a $k$-graphlet is any connected subgraph on $k$ vertices. Orthogonal to sampling instances of isomorphism types of subgraphs (e.g., triangles), graphlet sampling asks to sample a connected subgraph of a specific size. In an offline setting, Uniform Graphlet Sampling (UGS) solves this problem in $k^{O(k)} \log n$ expected time with $O(nk^2 \log k + m)$ preprocessing time. The authors in this paper propose a semi-streaming version of UGS that uses $O(\log n)$ passes for preprocessing, and can repeatedly sample $\tilde{\Theta}(n / k^{O(k)})$ graphlets uniformly at random using $O(k)$ passes. The running time of the preprocessing is essentially linear in its passes, and the sampling runs $\tilde{O}(n 2^k + mk)$ time.

UGS relies on a total ordering of the vertices so that for any two vertices $u > v$, the degree of $u$ is at most the degree of $v$ in the graph induced by all $w$ such that $w \geq v$. In a streaming setting, the challenge of constructing such an ordering is storing a representation of these induced graphs is to costly. To remedy this problem, the authors relax the guarantees of the ordering by scaling the degree of $u$ by some parameter $\vartheta \in (0,1]$. The authors show that one can construct such an ordering by bucketizing the vertices by degree and carfully partitioning the buckets into sparse induced subgraphs, which correspond to intervals in this ordering. The second building block of the algorithm is a rejection sampling scheme that can be run in parallel on top of the ordering.

The authors develop a heuristic version of their algorithm and evaluate both versions experimentally. It shows that the heuristic version seems to perform significantly better on sparse graphs. On large real-world sparse graphs, the heuristic version has an much larger heuristic success probability. It uses roughly 30 passes on the Friendster and Twitter graphs to sample 100 graphlets.

**Strengths:**

In comparison to previous work, the sample distribution is guaranteed to be exactly uniform over all graphlets.

Overall, the approach of carefully approximating the ordering is appealing and not completely trivial.

**Weaknesses:**

While it can provide strong guarantees for dense graphs, the semi-streaming model has limited applications for sparse graphs. It's not entirely clear whether $\Omega(\log n + k)$ semi-streaming passes on a sparse graph have as many applications as in the dense graph model.

**Questions:**

Could you elaborate on applications where $\log n$ passes are clearly superior to the potentially small increase of footprint for storing the whole graph in memory?

**Limitations:**

/

---

> ### Author Rebuttal · Authors · 2024-08-07
>
> We would like to thank the reviewer for the comments, which we would like to address as follows.
>
> [Response to Weakness]
>
> Our approach can deal with both sparse and dense graphs. The same theoretical guarantees work for sparse graphs and a good performance on sparse graphs by our approach is shown in the experiments. It is possible that there might be specialized algorithms for sparse graphs but our approach aims to provide a general solution to the problem, i.e. to both sparse and dense graphs.
>
> [Response to Hypothetical Application Scenario.]
>
> The input graph can be very dense, in some cases of interest. For example, we may need to study the similarity graph between a large number of pictures, say 1,000,000. Here, each vertex represents a picture, with each edge between two vertices indicating that the corresponding pictures are “similar”. Depending on the similarity measure at hand (e.g. cosine similarity larger than a given threshold), the resulting graph can be very dense (over 10^10 edges), in which case storing the whole input graph in memory would not be feasible. This motivates the development of a streaming algorithm, requiring a “small” amount of memory.

---

> > ### Comment · Reviewer_T1eM · 2024-08-08
> >
> > Thank you for your response! I agree with the application scenario, and actually I wouldn't challenge the advantages of semi-streaming for dense graphs. Am I further reading and combining the two parts of the answer correctly if I understand the following: The theoretical guarantees of the algorithm provide no clear benefit for sparse graphs over an offline algorithm that stores the graphs in memory. The point you make is that one may still use your algorithm for sparse graphs without worrying, though, as the experiments show good performance for the sparse instance.

---

> ### Author Response · Authors · 2024-08-09
>
> You are correct that if the sparse graphs can be fit into the memory, then the semi-streaming model would not provide any benefit.  However, in practice the constants matter; for instance, if the memory is sufficient only to store half of the graph, then our approach will still make sense.

---

> > ### Comment · Reviewer_T1eM · 2024-08-14
> >
> > Yes, I agree, in practice there's a sharp instead of an asymptotic threshold where the input doesn't fit into memory any more, and theoretical graph models like sparse or dense graphs are less important. Thank you for taking the time to answer the rebuttal!

---

### Official Review · Reviewer_NpXf · 2024-07-12

**Soundness:** 3
**Presentation:** 2
**Contribution:** 3
**Rating:** 7
**Confidence:** 3

**Summary:**

This paper studies uniform sampling of k-graphlets. Authors exttend UGS to the semi-streaming setting, and propose an time and space efficient framework that samples uniform k-grpahlets w.h.p. Authors also provide a lower bound on the memory requirement of any streaming algorithm for uniform k-graphlet sampling via the randomized communication model.Finally they conduct experiments on large scale dataset to verify the effectiveness.

**Strengths:**

Starting from UGS, Authors provide a framework to efficiently and uniformly sample k-graphlets. To achieve this, they defined DD order and propose an efficient semi-streaming algorithm that approximate the order. Then a distribution $p$ is computed based on the approxiamted DD order, and nodes are sampled from $p$. Finally a graphlet is randomly expanded from the initial node and accepted with probability. While the peel part of algorithm 1 is similar to the streaming algorithm for core order, it is interesting to see the shave operation can improve the approximation.

Authors prove the memory lower bound for the problem and thus shows their algorithm is near  optimal.

Authors conduct experiments on large graphs

**Weaknesses:**

In the related work, More sentences can be used to describe the UGS algorithm, especially the second phase. Currently there is no connection between the topological order and the sampling phase. Also random communication complexity can be introduced as it's the tool used for the lower bound proof.

Enlarge the font size of Figure legends.

**Questions:**

Given any G, there should always be a node v at the end of the order, such that G(v) is only this node itself right? In that sense $d(v|G(v))=0$ that violates def 2.1.

I agree with authors that DD order is not reverse of core order,

---

> ### Author Rebuttal · Authors · 2024-08-07
>
> We would like to thank the reviewer for the comments, which we would like to address as follows.
>
> [Response to Weaknesses]
>
> An efficient UGS sampling algorithm in the standard RAM model is described in the third paragraph of the introduction, which mentions the high level ideas of how topological order is used in uniform sampling.  We will clarify in the next version that the topological order is used to define the starting vertex of the sampling and compute the relevant probability.
>
> We will add a paragraph for the background of communication complexity lower bound in the related work.
>
> [Question: Given any G, there should always be a node v at the end of the order, such that G(v) is only this node itself right? In that sense d(v|G(v))=0 that violates def 2.1.]
>
> Definition 2.1 says that in an ordering, if d(v|G(v)) > 1, then some condition has to be satisfied. However, if d(v|G(v)) = 1 or 0, this means all the vertices in the remaining G(v) have degree 1 or 0, whose connected component has size at most 2.  Since the sampling problem is non-trivial only when k >= 3, in this case the ordering of the remaining vertices in G(v) is not important.

---

> > ### Comment · Reviewer_NpXf · 2024-08-13
> >
> > Thank you for your response!

---

### Decision · Program_Chairs · 2024-09-25

**Decision:**

Accept (poster)

**Comment:**

This paper gives an algorithm (and an almost matching lower bound) for the problem of sampling a connected k-vertex subgraph (graphlet) from a large graph in the semi-streaming model, where the algorithm is allowed to do several passes over the whole set of edges but can use only O(n log n) memory (where n denotes the number of nodes). The reviewers appreciate the interesting and nontrivial theoretical insights used to proved the main result, as well as the fact that it is supported by experiments. There are no major weaknesses that are not eventually addressed in the rebuttal. The only serious concern left is whether NeurIPS is the right venue for a graph mining paper. The authors mention several ML-related applications in their rebuttal, and I expect them to elaborate further on these applications in the camera-ready version.